# Efficient Speech Language Modeling via Energy Distance in Continuous Latent Space

**Zhengrui Ma** [1,2,3], **Yang Feng** [1,2] *, **Chenze Shao** [3], **Fandong Meng** [3], **Jie Zhou** [3], **Min Zhang** [4]

[1] Key Laboratory of Intelligent Information Processing
Institute of Computing Technology, Chinese Academy of Sciences
[2] University of Chinese Academy of Sciences
[3] Pattern Recognition Center, WeChat AI, Tencent Inc
[4] School of Future Science and Engineering, Soochow University
*: Corresponding author   {mazhengrui21b,fengyang}@ict.ac.cn

## Abstract

We introduce *SLED*, an alternative approach to speech language modeling by encoding speech waveforms into sequences of continuous latent representations and modeling them autoregressively using an energy distance objective. The energy distance offers an analytical measure of the distributional gap by contrasting simulated and target samples, enabling efficient training to capture the underlying continuous autoregressive distribution. By bypassing reliance on residual vector quantization, SLED avoids discretization errors and eliminates the need for the complicated hierarchical architectures common in existing speech language models. It simplifies the overall modeling pipeline while preserving the richness of speech information and maintaining inference efficiency. Empirical results demonstrate that SLED achieves strong performance in both zero-shot and streaming speech synthesis, showing its potential for broader applications in general-purpose speech language models. Demos and code are available at `https://github.com/ictnlp/SLED-TTS`.

## 1   Introduction

Text language modeling has achieved tremendous success in recent years. Works such as the GPT series [2, 50] have demonstrated that by modeling text sequences in an autoregressive manner and pretraining on large corpora, language models acquire impressive in-context learning abilities and can perform nearly any downstream natural language processing task. This remarkable potential of autoregressive language modeling has inspired researchers to explore whether speech audio, which also possesses a linear structure, can be modeled in a similar fashion.

Unlike text sequences composed of discrete tokens from a finite vocabulary, speech audio is often represented as a lengthy sequence of sampling points usually stored as integers or floating-point values within a range. This fundamental difference poses significant challenges for autoregressive speech modeling in a manner analogous to text. Text language models typically rely on a finite vocabulary to acquire a categorical distribution at each step, which is essential for stable training via cross-entropy loss and efficient ancestral sampling during inference. To adapt autoregressive modeling to speech, researchers have explored discretizing speech into sequences of discrete tokens using external quantization modules. Examples include discretization on SSL-learned features [24] in [33, 85, 49], discretization on ASR-supervised features in [7, 8, 83, 31] and reconstruction-oriented codec [82, 9, 32, 86, 80, 51, 25] in [1, 72, 88, 10, 81, 38]. However, discretization inevitably creates an information bottleneck, potentially losing the rich details in the raw waveform and thus degrading the reconstruction quality derived from the discretized tokens. One popular approach to mitigating this information loss is residual vector quantization [RVQ; 70, 82], which converts raw waveforms into

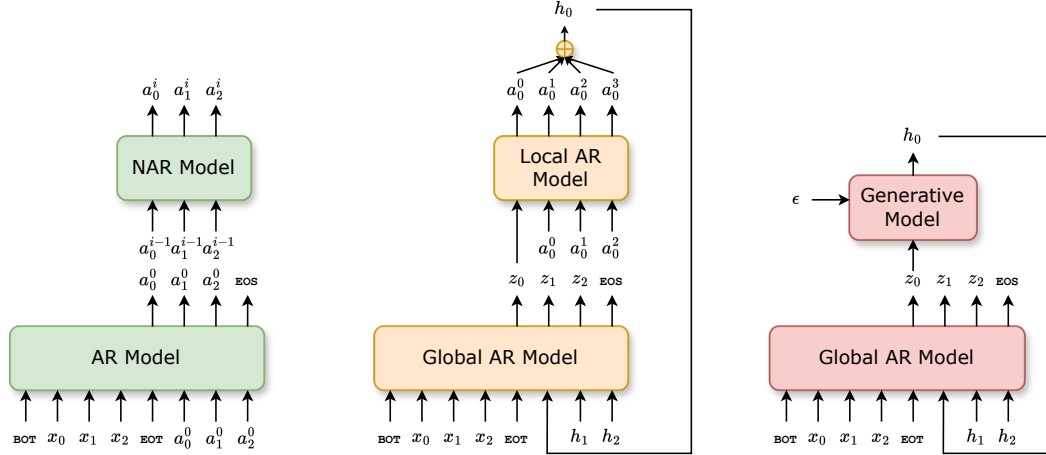

Figure 1: **Different speech language modeling approaches**. *Left*: VALL-E-style hierarchical architecture for RVQ token sequences. *Middle*: RQ-Transformer-style hierarchical architecture for RVQ token sequences. *Right*: Architecture for continuous token sequences.

multi-stream sequences of discrete tokens. However, RVQ introduces additional complexity, requiring hierarchical autoregressive architectures to effectively model multi-stream sequences [72, 77].

As an alternative to discretizing speech, researchers are starting to explore whether speech language modeling can be performed in a continuous latent space [46, 69, 39]. This approach offers several clear advantages. Firstly, it eliminates the need for discretization, enabling speech modeling in a latent space with minimal information loss. Secondly, it removes the reliance on hierarchical architecture required for modeling multi-stream sequences in RVQ, significantly reducing modeling complexity. Similar to discrete autoregressive modeling, continuous autoregressive modeling requires capturing a per-step distribution, but within a continuous space. Unlike the discrete case, where a categorical distribution can be constructed using the softmax function, continuous space modeling relies on a conditional probabilistic generative module to fulfill this role [68]. This module should be conditioned on features extracted by an autoregressive network, which captures the dependencies between the current step and the preceding ones. This raises an important question: **How should we construct such a generative module for speech language modeling in continuous latent space**? Ideally, we want the conditional generative module to function like the softmax in the discrete case—it should be lightweight, expressive, training-stable and inference-efficient. **In speech language modeling, sampling efficiency becomes particularly important** as the latent sequences of speech are typically lengthy, and each step in autoregressive generation involves a sampling process.

In this research, we introduce an approach to speech language modeling in continuous latent space using energy distance (SLED). We model the per-step continuous distribution with a lightweight implicit conditional generative module that takes features encoded with autoregressive dependencies and random noise as inputs [37, 60]. To measure the discrepancy between the underlying data distribution and the model's distribution, we employ a specialized form of maximum mean discrepancy [MMD; 17], known as generalized energy distance [GED; 42]. By selecting an appropriate distance function, the resulting metric becomes a strictly proper scoring rule [15, 59], enabling efficient training of the model to capture the autoregressive continuous distribution. Our approach eliminates the need for iterative sampling at each autoregressive step [37, 69] or post-AR refinement [46], significantly improving efficiency in modeling lengthy speech sequences while maintaining sufficient capacity. We demonstrate the effectiveness of our method in zero-shot and streaming speech synthesis, showing its strong potential for application in general-purpose speech language models.

## 2 Maximum Mean Discrepancy and Generalized Energy Distance

Integral probability metrics [47] are types of distance functions between probability distributions over $\mathbb{R}^n$. Let $\mathcal{F}$ be a class of real-valued functions on $\mathbb{R}^n$, integral probability metric is defined as:

$$D_{\mathcal{F}}(p, q) = \sup_{f \in \mathcal{F}}[\mathbb{E}_{\boldsymbol{x} \sim p(\boldsymbol{x})}[f(\boldsymbol{x})] - \mathbb{E}_{\boldsymbol{y} \sim q(\boldsymbol{y})}[f(\boldsymbol{y})]], \tag{1}$$

where $p$ and $q$ can be model and data distributions. Due to the richness of function class $\mathcal{F}$, it is often not practical to estimate $D_{\mathcal{F}}(p, q)$ with finite samples. *Maximum mean discrepancy* [MMD; 17] solves this problem by restricting the function class $\mathcal{F}$ to be the unit ball in a reproducing kernel Hilbert space $\mathcal{H}$:

$$\text{MMD}(p, q) = \sup_{\|f\|_{\mathcal{H}} \leq 1} [\mathbb{E}_{\boldsymbol{x} \sim p(\boldsymbol{x})}[f(\boldsymbol{x})] - \mathbb{E}_{\boldsymbol{y} \sim q(\boldsymbol{y})}[f(\boldsymbol{y})]]. \tag{2}$$

Assuming real-valued function $k(\boldsymbol{x}, \boldsymbol{y})$ is symmetric and positive definite, the kernel $k$ defines a reproducing kernel Hilbert space $\mathcal{H}$ such that every critic function $f \in \mathcal{H}$ can be expressed as $f(\boldsymbol{x}) = \langle f, k(\boldsymbol{x}, \cdot) \rangle_{\mathcal{H}}$. This notion can be extended to the embedding of a probability distribution by defining $\mu_p \in \mathcal{H}$ such that $\mathbb{E}_{\boldsymbol{x} \sim p(\boldsymbol{x})}[f(\boldsymbol{x})] = \langle f, \mu_p \rangle_{\mathcal{H}}$ for all $f \in \mathcal{H}$. Gretton et al. [17] shows that, if the condition for the existence of $\mu_p$ is satisfied, $\mu_p = \mathbb{E}_{\boldsymbol{x} \sim p(\boldsymbol{x})}[k(\boldsymbol{x}, \cdot)]$ and MMD can be expressed as the distance between mean embeddings:

$$\text{MMD}_k^2(p, q) = \|\mu_p - \mu_q\|_{\mathcal{H}}^2 = \mathbb{E}_{\substack{\boldsymbol{x}, \boldsymbol{x}' \sim p \\ \boldsymbol{y}, \boldsymbol{y}' \sim q}}[k(\boldsymbol{x}, \boldsymbol{x}') + k(\boldsymbol{y}, \boldsymbol{y}') - 2k(\boldsymbol{x}, \boldsymbol{y})], \tag{3}$$

where $\boldsymbol{x}, \boldsymbol{x}'$ and $\boldsymbol{y}, \boldsymbol{y}'$ are independent samples from $p$ and $q$.

Müller [47] has shown that $D_{\mathcal{F}}$ is a *pseudometric* for any choice of $\mathcal{F}$, which satisfies all the properties of a *metric*[1] except that there can be $D_{\mathcal{F}}(p, q) = 0$ when $p$ is not identical to $q$. However, $\text{MMD}(p, q)$ is a metric when the mean embedding $\mu_p$ is injective [17]. Kernels that satisfy this condition are referred to as *characteristic kernels* [14]. By selecting a characteristic kernel function $k(\boldsymbol{x}, \boldsymbol{y})$, $\text{MMD}(p, q)$ becomes a strictly proper scoring rule [15] and can be effectively used to train implicit probabilistic generative models.

In an alternative view, Lyons [42] defines *generalized energy distance* (GED) between two probability distributions on metric spaces $(\mathbb{R}^n, d)$ as:

$$\text{GED}_d^2(p, q) = \mathbb{E}_{\substack{\boldsymbol{x}, \boldsymbol{x}' \sim p \\ \boldsymbol{y}, \boldsymbol{y}' \sim q}}[2d(\boldsymbol{x}, \boldsymbol{y}) - d(\boldsymbol{x}, \boldsymbol{x}') - d(\boldsymbol{y}, \boldsymbol{y}')]. \tag{4}$$

Several works [15, 42] have pointed out that if the distance function $d$ is of *negative type* or *conditionally negative definite*, then $\text{GED}^2(p, q) \geq 0$, and it forms a pseudometric between distributions. Actually, GED is a special case of MMD. Sejdinovic et al. [57] shows that any *nondegenerate* kernel $k$ on $\mathbb{R}^n$ defines a valid *semimetric* $d$ of negative type by:[2]

$$d(\boldsymbol{x}, \boldsymbol{y}) = k(\boldsymbol{x}, \boldsymbol{x}) + k(\boldsymbol{y}, \boldsymbol{y}) - 2k(\boldsymbol{x}, \boldsymbol{y}), \tag{5}$$

and any semimetric $d$ of negative type can be generated by at least one of its induced kernels:

$$k(\boldsymbol{x}, \boldsymbol{y}) = d(\boldsymbol{x}, \boldsymbol{z}) + d(\boldsymbol{y}, \boldsymbol{z}) - 2d(\boldsymbol{x}, \boldsymbol{y}), \tag{6}$$

where $\boldsymbol{z}$ can be any point on $\mathbb{R}^n$. $\text{GED}_d^2$ is equivalent to the $\text{MMD}_k^2$ associated to a kernel $k$ that generates $d$ [57]. Therefore, one can alternatively use GED as a criterion for training implicit generative models, as long as the distance function $d$ is chosen appropriately.

## 3 Approach

### 3.1 Language Modeling in Continuous Latent Space

Prior approaches to speech language modeling have commonly depended on an external quantization module, which discretizes speech waveform into a sequence of tokens from a finite vocabulary for autoregressive modeling in a manner analogous to text. The generated speech token sequence can be resynthesized into a waveform using either the decoder of the quantization model or an externally trained vocoder [53, 63]. In this research, we explore encoding speech waveforms into sequences of continuous representation and performing autoregressive modeling within this continuous latent space. Given an audio sample $\boldsymbol{x} \in \mathbb{R}^{Tf}$, we encode it into a sequence of continuous representation: $\boldsymbol{h} = \text{Enc}(\boldsymbol{x})$ $\boldsymbol{h} \in \mathbb{R}^{Tf_h \times n}$, where $T$ is the duration of the audio, and $f$ and $f_h$ denote the sample rates of the audio waveform and the latent sequence. Typically, $f$ ranges from hundreds of thousands

---

[1]A metric satisfies the following four properties: non-negativity, symmetry, triangle inequality and identity of indiscernibles.

[2]The definitions of nondegenerate kernels and semimetrics can be found in Sejdinovic et al. [57].

to several hundred thousand, while $f_h$ ranges from a few to tens. The model aims to capture the distribution of the continuous vector $h_t$, conditioned on all previously generated vectors $h_{<t}$:

$$p(\boldsymbol{h}) = \prod_{t=0}^{Tf_h - 1} p(\boldsymbol{h}_t | \boldsymbol{h}_{<t}). \tag{7}$$

In the discrete domain, autoregressive models can use *softmax* function to model the distribution of each step over the vocabulary space. However, autoregressive modeling in the continuous domain is more complex. It requires modeling the distribution in the $\mathbb{R}^n$ space for each step, conditioned on the outputs from previous steps:

$$\boldsymbol{z}_t = \psi(\boldsymbol{h}_{<t}; \theta), \ \hat{\boldsymbol{h}}_t \sim p_g(\boldsymbol{h}_t | \boldsymbol{z}_t; \phi) \tag{8}$$

where $\psi$ can be any autoregressive network (e.g., decoder-only Transformer [71]) and $g$ can be any conditional generative module on $\mathbb{R}^n$.

## 3.2 Per-token Generative Modeling via Energy Distance

As discussed earlier, we perform autoregressive modeling in the continuous latent space using a conditional generative module $g$ per step. The model $g$ is responsible for modeling the distribution of the current step $h_t$ based on the condition representation $z_t$, which incorporates autoregressive dependencies. Ideally, any conditional generative model can serve as $g$. However, in the context of speech language modeling, $g$ needs to meet certain requirements. **Modeling Capability**: Considering the diversity of speech audio, $g$ requires a strong capacity to capture the distribution at each step. Analytically tractable distributions, such as Gaussian distributions, may not have sufficient capability. **Sampling Efficiency**: Speech audio has a high sampling rate. Even though this rate is significantly reduced after encoding into the continuous latent space, the sequences remain lengthy. Given that each step of autoregressive decoding requires a sampling operation from $g$, we aim for $g$ to have high sampling efficiency, akin to categorical sampling over vocabulary. If we model $g$ using methods such as diffusion [23], which require multiple iterations during sampling, it could significantly increase the latency of autoregressive decoding. **Training Stability**: Similar to training discrete autoregressive models using cross-entropy loss, we aim to train both the conditional generative module $g$ and the main autoregressive network $\psi$ simultaneously using a simple and stable training algorithm. Under such constraints, the GAN-style training paradigm [16] may not be appropriate.

Based on the above analysis, we construct $g$ as a lightweight multi-layer perceptron that takes the condition vector $z_t$ and noise vector $\epsilon$ as inputs, mapping them to the continuous latent space of speech, $\mathbb{R}^n$:

$$\boldsymbol{h}_t = g(\boldsymbol{z}_t, \epsilon; \phi). \tag{9}$$

This implicitly defines a per-step distribution $p_g(\boldsymbol{h}_t | \boldsymbol{z}_t)$ over $\mathbb{R}^n$. The sampling process involves simply sampling a noise vector $\epsilon_t$ and passing it through $g$ along with the condition $z_t$. In the network $g$, we use the AdaLN module [52] to integrate noise into the hidden state of the condition vector $z_t$ to introduce randomness. Rather than directly learning dimension-wise scale and shift parameters in the layer normalization module for $z_t$, we treat them as random perturbations for sampling [60]. The scale and shift values are predicted by applying a linear transformation to the input noise $\epsilon_t$.

To train the implicit conditional generative module $g$ and the autoregressive network $\psi$ simultaneously, we use a specialized form of maximum mean discrepancy, namely *energy distance*, as the metric between the distributions of the data and model. The training is performed by minimizing the energy distance between the simulated and the ground truth latent representation per step, with respect to the parameters of both $g$ and $\psi$. Since the data distribution remains fixed during optimization, terms in Eq. 4 that do not depend on the model parameters can be discarded, reducing the training loss to:

$$\mathcal{L}_{\text{GED}} = \sum_t \mathbb{E}_{\boldsymbol{h}_t, \boldsymbol{h}_t'} [2d(\boldsymbol{h}_t, \boldsymbol{h}_t^*) - d(\boldsymbol{h}_t, \boldsymbol{h}_t')], \tag{10}$$

where $\boldsymbol{h}^*$ is the target latent waveform representation, and $\boldsymbol{h}_t$, $\boldsymbol{h}_t'$ are independent samples from $p_g(\boldsymbol{h}_t | \boldsymbol{z}_t)$. As discussed in Sec. 2, $\mathcal{L}_{\text{GED}}$ is a strictly proper scoring rule, provided that the kernel function corresponding to the distance function $d$ is characteristic. [15, 66] suggests that $d(\boldsymbol{x}, \boldsymbol{y}) = \|\boldsymbol{x} - \boldsymbol{y}\|_2^\beta$ satisfies the condition when $\beta \in (0, 2)$. In our experiments, we set $\beta = 1$:

$$\mathcal{L}_{\text{GED}} = \sum_t \mathbb{E}_{\boldsymbol{h}_t, \boldsymbol{h}_t'} [2\|\boldsymbol{h}_t - \boldsymbol{h}_t^*\|_2^1 - \|\boldsymbol{h}_t - \boldsymbol{h}_t'\|_2^1], \tag{11}$$

It is worth noting that Eq. 11 is very similar to the root mean squared error, with the addition of the repulsive term $\mathbb{E}[\|\boldsymbol{h}_t - \boldsymbol{h}_t'\|_2^1]$. This repulsive term is crucial for making the loss a proper learning objective. In contrast, RMSE is only a regression loss, which does not well capture the differences between the distributions. Using RMSE loss as the objective can cause the model to fail to fit the data distribution [18]. We demonstrate this in Sec. 6.1.

SLED performs autoregressive modeling in the continuous latent space, thus it cannot determine when to stop by predicting the EOS token. To address this issue, we adopt the approach from [46] and introduce a binary classification head to decide whether the output should terminate. We directly project the output of the autoregressive network $\boldsymbol{z}_t$ to a scalar, followed by a sigmoid function, and interpret the result as the probability of whether the output should stop at this step.

### 3.3 Classifier-free Guidance

Continuous latent generation exhibits higher noise levels than its discrete counterpart. To improve the generation quality and the alignment with prompts, we adopt the classifier-free guidance technique [CFG; 22, 60] during inference. At each step of autoregressive generation, we perform an additional forward pass through $\psi$, during which the prompt text is masked out, to obtain $\boldsymbol{z}_t'$. We then linearly interpolate $\boldsymbol{z}_t$ with $\boldsymbol{z}_t'$ and use the result as input to the per-token generative module $g$:

$$\boldsymbol{z}_t^{\text{cfg}} = \boldsymbol{z}_t' + \lambda(\boldsymbol{z}_t - \boldsymbol{z}_t'), \; \boldsymbol{h}_t^{\text{cfg}} = g(\boldsymbol{z}_t^{\text{cfg}}, \epsilon; \phi), \tag{12}$$

where $\lambda$ is the hyperparameter to control the strength of the guidance and it reverts to naïve conditional generation when $\lambda = 1.0$. During training, we randomly mask out the text prompt with a probability of 0.1. The EOT token is preserved during masking, as it also serves as the token indicating the beginning of the speech.

### 3.4 Streaming Inference

One appealing feature of our approach is that it is a *purely* autoregressive model and does not require any post-processing module to refine the outputs at earlier positions after the entire autoregressive generation is complete. This desirable property allows our autoregressive modeling approach to support ***streaming generation*** [43, 8, 78, 44]. Here, streaming generation has two levels of meaning: 1) After each autoregressive step generates a latent vector, it can immediately be used for waveform synthesis. This capability relies on the model's autoregressive generation not requiring any post-processing steps, and the decoder that converts latent vectors to waveforms supporting streaming. 2) Building on this, we aim for the model to begin generating even before the full prompt text is provided. This incremental generation feature is particularly useful to reduce the response latency of a GPT-4o-like speech interaction system [12, 74, 3, 76, 13], where the model serves as a TTS module following a text LLM.

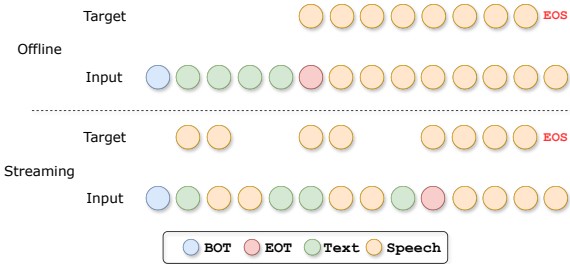

Figure 2: Illustration of our streaming inference mechanism. Text and speech tokens are interleaved based on a predefined ratio, and the loss is computed only at positions where targets are shown.

Modern audio codecs typically have a streaming decoder [9], so the first level of streaming generation is naturally supported. To achieve incremental speech synthesis, we alter the order of the text and speech positions within the sequence during autoregressive modeling [8]. As illustrated in Figure 2, we interleave text and speech positions in an $n : m$ ratio. This design allows the generation of $m$ speech vectors for every $n$ text tokens received, enabling incremental text-to-speech synthesis. An EOT token is used to indicate the end of the input text stream. Afterwards, the speech vectors are generated in the standard autoregressive way until a stop decision is made by the binary classification head.

We train the model to generate target vectors and predict whether to stop only at the input positions that require generation during inference. We do not require the model to predict any filling tokens when the next step is a text token according to the interleaving ratio. We also adopt the classifier-free guidance outlined in Sec. 3.3 in streaming inference. The model performs unconditional speech generation when the text stream requires masking, with EOT serving as the beginning.

# 4  Related Work

Given great success of autoregressive language modeling approach in text generation, researchers have started exploring whether speech sequences can be modeled in a similar way. These studies begin with speech synthesis as a foundational task [33, 1, 72], eventually extending to the development of general-purpose speech language models [48, 85, 10, 83, 38, 31]. [33] was the first to propose discretizing self-supervised representations of speech as semantic tokens and to perform autoregressive speech generation. The method of learning semantic tokens has since evolved to quantize ASR-supervised representations [7, 8]. Due to information loss, waveforms reconstructed from coarse semantic tokens are often unsatisfactory, necessitating modeling with acoustic tokens produced by reconstruction-oriented audio codecs [82, 9, 32, 86, 80, 51, 25]. Since the acoustic tokens generated by codecs often consist of multiple sequences from residual quantization, a hierarchical architecture is required for autoregressive speech modeling. [72] utilize an autoregressive model to generate the sequence of tokens from the first codebook, followed by a non-autoregressive model to generate tokens from the residual codebooks. To mitigate the latency introduced by post-AR refinement, [77, 88] apply the RQ-Transformer [35], which uses a smaller nested Transformer to model tokens from multiple codebooks within a single autoregressive step. Nonetheless, modeling multi-stream acoustic tokens introduces additional complexity. Inspired by advancements in image generation [68, 37], researchers have begun exploring performing speech language modeling in the continuous latent space. [46] use mel-spectrograms as the latent representation and model the per-step distribution with a regression loss. However, using a regression loss fails to capture the underlying distribution, requiring a post-AR refinement network and handcrafted repetition loss. [69] adopt a per-token diffusion loss [37], which requires iterative sampling for each autoregressive step, leading to significant inference latency. [73] models each step via flow matching [40], which also encounters similar problems. To this end, [26] proposes a patch-based continuous autoregressive framework that couples a language model for inter-patch prediction with a DiT [52] for intra-patch generation, thereby reducing the number of sampling steps. [87, 39] model the data at each autoregressive step with a Gaussian distribution or GMM and employ a VAE to obtain the target distribution parameters for each step. These approaches impose overly strong constraints on the latent space, limiting the model's expressive capacity.

# 5  Experiments

## 5.1  Training Datasets

We train SLED on the large-scale Libriheavy [29] dataset. LibriHeavy contains approximately 50,000 hours of speech from 6,736 speakers, deriving from audiobooks from the LibriVox project. A BPE tokenizer [58] with a vocabulary size of 16,384 is applied for text.

## 5.2  Experimental Settings

**Continuous Representation**  We employ Encodec [9] to extract continuous latent representations. For each frame, we sum the multi-stream token embeddings—drawn from eight codebooks—to form its continuous vector, ensuring minimal information loss. **Although a plain autoencoder can produce continuous latents, we find that enforcing a quantization-regularized or KL-regularized latent space substantially benefits downstream language modeling.** This finding is in line with the standard practice in latent diffusion models [56]. Considering the representative discrete modeling approach VALL-E [72] also employs Encodec as its tokenizer, this setup also enables a fair comparison between continuous and discrete autoregressive modeling while keeping the latent encoder consistent. The resulting continuous vector sequences have a sampling rate of 75Hz and a latent dimensionality of 128.

**Model Configurations**  The autoregressive network incorporates 12 Transformer layers [71]. Instead of using standard Transformer layers, we employ LLaMA [67] layers, which apply pre-normalization

Table 1: Performance comparison on *3s Prefix as Prompt* and *Reference Utterance as Prompt* zero-shot speech synthesis tasks. WER-C (%) represents the results using the Conformer-Transducer, whereas WER-H (%) indicates the results with the HuBERT-Large. *: WER obtained by Whisper.

| System | #Params | 3s Prefix as Prompt | | | Reference Utterance as Prompt | | |
|---|---|---|---|---|---|---|---|
| | | WER-C | WER-H | SIM | WER-C | WER-H | SIM |
| Ground Truth | - | 1.78 | 2.15 | 0.668 | 1.78 | 2.15 | 0.778 |
| Ground Truth (Encodec) | - | 1.79 | 2.30 | 0.606 | 1.79 | 2.30 | 0.738 |
| *Traditional TTS* | | | | | | | |
| MaskGCT [75] | 1.0B | - | - | - | - | 2.63 | 0.687 |
| F5-TTS [6] | 0.3B | - | - | - | - | 2.42* | 0.66 |
| MegaTTS 3 [27] | 0.3B | - | - | - | - | 1.82 | 0.78 |
| *Discrete-LM-based Approaches* | | | | | | | |
| VALL-E [72] | 0.4B | - | 3.8 | 0.508 | - | 5.9 | 0.580 |
| VALL-E 2 [5] | 0.4B | 1.6 | 2.32 | 0.529 | 1.5 | 2.44 | 0.678 |
| ELLA-V [64] | 0.4B | 2.10 | 2.91 | 0.340 | 7.15 | 8.90 | 0.331 |
| VALL-E R [20] | 0.4B | 1.58 | 2.32 | 0.397 | 3.18 | 3.97 | 0.395 |
| Llasa [81] | 8B | - | 1.49 | 0.740 | - | - | - |
| *Continuous-LM-based Approaches* | | | | | | | |
| CLaM-TTS [30] | - | - | 2.36 | 0.513 | - | 5.11 | 0.538 |
| MELLE [46] | 0.2B | 1.47 | 1.98 | 0.539 | 1.47 | 2.10 | 0.664 |
| FELLE [73] | 0.2B | 1.53 | 2.27 | 0.539 | 2.20 | 2.89 | 0.654 |
| SLED | 0.2B | 1.59 | 1.99 | 0.515 | 1.51 | 1.97 | 0.664 |

Table 2: Performance comparison of *Offline* and *Streaming Inference*.

| Mode | $n:m$ | WER-C | DNSMOS |
|---|---|---|---|
| GT | – | 1.79 | 3.89 |
| Offline | – | 1.67 | 3.58 |
| with Prompt | – | 1.51 | 3.61 |
| Streaming | 5:20 | 2.18 | 3.59 |
| Streaming | 5:45 | 2.20 | 3.54 |

Table 3: Performance comparison using Energy Distance vs. Mean Squared Error as the objective in *3s Prefix as Prompt* experiments.

| Objective | WER-C |
|---|---|
| Root Mean Squared Error | 40.60 |
| Energy Distance | 1.59 |

using RMSNorm [84], SwiGLU activation function [62], and rotary positional embeddings [65]. Each layer has 16 attention heads and an embedding dimension of 1,024. We set the FFN hidden dimension to 2,752 to match the number of parameters of a standard Transformer with an FFN hidden dimension of 4,096. The dropout rate is set at 0.1. The input latent continuous vectors are projected to the embedding dimensionality using a linear layer, followed by a layer normalization to ensure stability. The lightweight conditional generative module consists of six residual blocks, each featuring an AdaLN module [52] to modulate the hidden states with noise, followed by a two-layer MLP and residual connections. The hidden dimensionality of this generative module is set to 1024, and a linear layer is used at the top to project the output back to the target dimensionality. The default value of $\lambda$ in classifier-free guidance is set to 2.0.

**Training Details** We train the model with a batch size of 512 for 300,000 steps using BF16. We optimize the model with AdamW [41], configured with a learning rate of 5e-4, weight decay of 0.01, $\beta_1 = 0.9$, $\beta_2 = 0.999$ and $\epsilon = 1 \times 10^{-8}$. The learning rate follows a linear decay, warming up to its peak value during the first 32,000 steps. A maximum gradient norm clip of 1.0 is applied.

## 5.3 Evaluation Settings

We evaluate zero-shot speech synthesis performance using LibriSpeech *test-clean* set, ensuring that none of the test speakers are included in the training data. Following [72, 46], we use samples with durations between 4 and 10 seconds, resulting in a 2.2-hour subset comprising 1,234 samples and 40 unique speakers. Since the LibriSpeech consists of case-insensitive text without punctuation, while

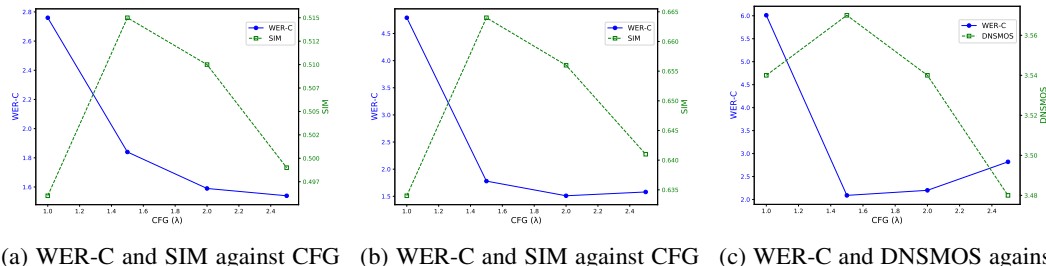

(a) WER-C and SIM against CFG in *3s Prefix as Prompt*.

(b) WER-C and SIM against CFG in *Ref Utterance as Prompt*.

(c) WER-C and DNSMOS against CFG in *Streaming Inference*.

Figure 3: Evaluation of classifier-free guidance effects across generation settings.

the expected input for models trained on LibriHeavy is unnormalized, we use the *test-clean* set of LibriSpeech-PC [45] to evaluate SLED. The LibriSpeech-PC is nearly identical to LibriSpeech, but its text has been restored to include capitalization and punctuation.[3] The evaluation is conducted under two settings: 1) *3s Prefix as Prompt*: The model is provided with the text transcription and the first 3 seconds of the utterance as the prompt, and it is expected to perform speech continuation. 2) *Reference Utterance as Prompt*: The model is given a reference utterance from the same speaker along with its transcription as the prompt. Using the text of the target utterance, the model is expected to synthesize speech that retains the characteristics of the reference speaker. For this experiment, we follow the configuration outlined in [5]. We first filter the samples in LibriSpeech test-clean set based on length and order them by sample ID. For each speaker, the $i$-th speech sample is synthesized using the $(i-1)$-th sample as the prompt, while the first sample is synthesized using the last sample from the same speaker as the prompt. For *Offline* and *Streaming Inference* experiments, we follow previous data settings and no prompt speech is provided. We primarily use the following metrics to assess the intelligibility and in-context learning capability.

**Word Error Rate (WER)** To evaluate the intelligibility of the synthesized speech, we perform speech recognition on the generated audio and calculate the WER against the target text. We utilize two ASR models: Conformer-Transducer [19] and CTC-ASR-tuned HuBERT-Large [24].

**Speaker Similarity (SIM)** We evaluate the in-context learning capability by measuring speaker similarity between the reference and the generated speech. We extract speaker embeddings using WavLM-TDNN [4] and compute their cosine similarity.

**Predicted Mean Opinion Score (DNSMOS)** We evaluate the overall quality of the generated speech using the DNSMOS score [54, 55], which ranges from 1 to 5, with higher values indicating better quality. We use a model trained with ground truth human ratings obtained using ITU-T P.808.

### 5.4 Main Results

**Zero-shot TTS** Table 1 presents the results of zero-shot TTS. In both the speech continuation and reference utterance prompting tasks, SLED achieves a word accuracy surpassing the ground truth (1.59/1.51 vs. 1.78 in WER-C), demonstrating its modeling ability. Interestingly, we found that using a reference utterance as a prompt results in more accurate synthesis compared to using the prefix, showing the ability of in-context learning. Comparisons of SLED performance across different data scales are provided in App. A. We found that 1000 hours of speech is sufficient to achieve most of the generation and in-context learning capabilities. Further scaling the training data to 50,000 hours enhances these abilities even more. Notably, a comparison between SLED and VALL-E highlights key differences in autoregressive modeling approaches. Both models use Encodec as the latent encoder, but SLED performs modeling in the continuous domain, whereas VALL-E operates in the discrete domain. VALL-E-like hierarchical models require an extra non-autoregressive local model ($\approx$159M) to predict residual tokens. **In contrast, SLED relies on a lightweight generative module ($\approx$35M) for continuous vector generation, achieving better parameter efficiency.** Despite this, SLED outperforms VALL-E on all metrics. These results show the superiority of our continuous speech language modeling approach. **Compared with other continuous approaches, SLED achieves**

---

[3]The *test-clean* set of Librispeech-PC excludes a small number of low-quality samples (203 out of 2,620) from the original Librispeech *test-clean* set, thus the test subset after applying the same protocol contains only 1,154 samples.

**better or similar quality while enjoys the advantage in generation efficiency.** MELLE incorporates an NAR refinement, whereas SLED is purely autoregressive and can generate in a streaming manner. FELLE must perform multi-step integration iterations along a predefined ODE path at each decoding step, while SLED only requires a single forward pass through its generative module. Moreover, at roughly the same parameter scale, current speech language modeling approaches still exhibit a significant gap in voice cloning compared with traditional TTS models. However, Llasa [81] shows that scaling up can dramatically strengthen discrete autoregressive speech models, which inspires us to further explore the scaling capabilities of continuous autoregressive modeling approaches.

**Streaming Inference**

Table 2 presents a comparison of accuracy (WER-C) and perceived quality (DNSMOS) between speech produced via streaming inference and that generated offline. We vary the interleaving ratio between text and speech positions to analyze the impact of streaming policies. In our experiments, we compare two configurations: one generates 20 speech vectors (0.27 seconds of speech) for every 5 subwords received, while the other generates 45 speech vectors (0.6 seconds of speech) per 5 subwords. The latter configuration aligns with [8], which guarantees seamless streaming generation when serving as the speech synthesis module of a cascaded speech-LLM system. We find that the two interleaving ratio configurations yield comparable results in both word accuracy and perceived quality. **Moreover, the synthesized quality in streaming mode closely matches that of offline synthesis** (3.59/3.54 vs. 3.58 in DNSMOS), despite a slight drop in word accuracy (2.18/2.20 vs. 1.67 in WER-C), demonstrating the effectiveness of SLED for streaming tasks.

# 6 Analysis

## 6.1 Importance of Energy Distance

As discussed in Section 3.2, the learning objective of SLED (Eq. 11) closely resembles root mean squared error. By removing the repulsive term $\mathbb{E}[\|\boldsymbol{h}_t - \boldsymbol{h}'_t\|_2^1]$, the energy distance becomes equivalent to the RMSE, with random noise acting as a perturbation (which is the L2 part of the regression loss in [46]). Despite their apparent similarity, these two objectives exhibit distinct statistical properties. Energy distance measures the *distributional difference*, guiding the model to capture the underlying data distribution. In contrast, RMSE is only a *regression loss*. The importance of the repulsive term is demonstrated in Table 3. Removing this term results in a model failure, leading to a significantly worse word error rate. Listening to the generated samples, we observed that the model trained without the repulsive term failed to generate speech in most cases.

This makes us curious why MELLE [46] achieves strong performance using a regression loss. We note that [46] introduces a flux loss to suppress repetition, which measures the variation of the generated spectral sequence over time by taking the norm of the Mel-spectrogram differences between adjacent frames. [46] reports that generation quality degrades severely without it. **Although [46] frames the flux loss as a heuristic designed to curb repetition, we argue that its real efficacy stems from approximating the repulsive term in the energy distance**. Because consecutive frames in speech differ only slightly, they can be approximately regarded as two samples from the same time step. Under this view, MELLE's flux loss causes its training objective to approximate the energy distance, thereby enabling it to capture distributional differences to some extent.

## 6.2 Effects of Classifier-free Guidance

We investigate the impact of classifier-free guidance (CFG) across generation settings. As shown in Figure 3, without CFG ($\lambda = 1.0$ in Eq. 12), the synthesized speech exhibits poor word accuracy, with a particularly high WER-C of 6.01 in streaming inference. However, upon applying CFG, the WER-C values decrease dramatically. A $\lambda$ value of 1.5 quickly results in a substantial improvement, bringing the WER-C down to approximately 2.0 across all settings. Further tuning of $\lambda$ leads to even better word accuracy. However, in terms of timbre cloning and perceived speech quality, a moderate CFG setting ($\lambda = 1.5$) yields the most improvements. Further increasing $\lambda$ leads to performance degradation. Notably, speech quality in streaming inference declines to a level worse than that generated without CFG at $\lambda = 2.5$. This experiment demonstrates that classifier-free guidance significantly improves alignment between the text and synthesized speech while enhancing speech

quality only with a moderate $\lambda$ value. We recommend setting $\lambda = 2.0$ by default to achieve a balance between word accuracy and speech quality across all generation settings.

## 6.3 Efficiency Analysis

We view SLED and DiTAR [26] as complementary approaches to more efficient continuous autoregressive speech modeling. DiTAR reduces the number of sequential steps via a semi-autoregressive framework, yielding time complexity between fully autoregressive and fully non-autoregressive, whereas SLED accelerates sampling in the continuous latent space at each autoregressive step. We compare the computational efficiency of the two methods in Table 4. All the metrics are evaluated by inferring 10 seconds of audio.

Table 4: Comparison of decoding efficiency between SLED and DiTAR [26].

| Model | #Params | #Transformer Layer | RTF (bs=1) | FLOPs |
|---|---|---|---|---|
| SLED | 0.2B | 12 | 0.8 | 280G |
| DiTAR (patch size 4) | 0.6B | 6+36+6 | 0.66 | 2750G |

Although DiTAR's patching scheme reduces the number of sequential steps, the approach is hierarchical. Each step in DiTAR bears a heavy generation workload and therefore relies on a relatively large local decoder (DiT), in contrast to SLED's lightweight MLPs for per-step generation. This burden can erode the nominal benefits of patching and, by iterating within a local DiT, may also compromise global consistency; DiTAR paper reports that the local-DiT module must incorporate historical-patch dependencies to generate correctly, increasing complexity and narrowing the method's computational advantage. Empirically, SLED achieves a comparable RTF to DiTAR while using nearly 10× fewer FLOPs and roughly one-third the parameters.

## 7 Conclusion and Limitation

In this work, we propose a method for speech language modeling in continuous latent space using energy distance. This approach avoids the complex hierarchical architecture of traditional discrete models, while also preserving rich information in speech. This work has validated the effectiveness of this modeling approach in the speech synthesis task. In the future, we plan to extend it to general-purpose speech language models. Moreover, the latent encoder employed in the current work was originally designed solely for the audio codec; additionally training a continuous latent encoder specifically for this method should further enhance its performance.

## Acknowledgement

We gratefully acknowledge all the reviewers for their valuable comments and suggestions. This work was supported by the Natural Science Foundation of Beijing, China (Grant No. L257006).

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

## A   Results at Different Data Scales

Table 5: Comparison of model performance when trained on LibriSpeech versus LibriHeavy.

| System | 3s Prefix as Prompt | | | Reference Utterance as Prompt | | |
|---|---|---|---|---|---|---|
| | WER-C | WER-H | SIM | WER-C | WER-H | SIM |
| Ground Truth | 1.78 | 2.15 | 0.668 | 1.78 | 2.15 | 0.778 |
| Ground Truth (Encodec) | 1.79 | 2.30 | 0.606 | 1.79 | 2.30 | 0.738 |
| *Models trained on the 1,000-hour LibriSpeech* | | | | | | |
| SLED | 1.68 | 2.28 | 0.490 | 2.27 | 2.88 | 0.616 |
| *Models trained on the 50,000-hour LibriHeavy* | | | | | | |
| SLED | 1.59 | 1.99 | 0.515 | 1.51 | 1.97 | 0.664 |

## B   Why not Gaussian?

In Sec. 3.2, we claim Gaussian distribution does not have enough capacity to model the per-token continuous distribution in latent space. We have designed experiments to demonstrate this. Specifically, using a trained SLED (which we assume has successfully captured the underlying data distribution), we sampled 100 latent vectors from the first speech position and conducted a Shapiro-Wilk test [61] to examine whether each dimension follows a Gaussian distribution. The result showed that 0 out of 128 dimensions passed the normality test ($p > 0.05$), indicating that none of the dimensions individually follow a normal distribution.

## C   More Analysis of Inference Efficiency

Table 6: Inference latency and RTF of generating a 10s speech. *: We use teacher-forcing-style AR forward pass to estimate FLOPs in AR module.

| Model | AR Latency | Gen. Module Latency | NAR Latency | RTF | FLOPs* |
|---|---|---|---|---|---|
| VALL-E | 6.91 s | 0.16 s (Softmax) | 0.94 s | 0.8 | $8 \times 222.7G$ |
| SLED | 6.82 s | 1.23 s (MLP) | – | 0.8 | 280.05 G |

Table 7: RTF of SLED for different batch sizes.

| Batch Size | SLED's RTF |
|---|---|
| 1 | 0.8 |
| 16 | 0.055 |
| 64 | 0.027 |

As shown, SLED and discrete hierarchical models like VALL-E exhibit no significant difference in RTF. However, SLED largely reduces FLOPs compared with hierarchical models and supports streaming generation. The additional time spent by the MLP heads in continuous generation is offset by the efficiency gains from avoiding the decoding process required by local NAR models. Combined with its RTF being less than 1, SLED is well-suited for supporting real-time applications.

Moreover, while SLED's RTF may appear higher than NAR TTS models such as [6], we argue that comparing RTF between AR and iterative NAR with the batch size fixed to 1 is somewhat misleading. In fact, prior studies on NAR generation [21] have long advocated for evaluating AR and NAR models under larger batch sizes—up to the capacity supported by the GPU—or equivalently, under the same computational budget (FLOPs). In fact, we observe that increasing the batch size significantly reduces SLED's RTF.

## D   Broader Impact

The speech language modeling method presented in this paper enables voice cloning, and the experimental results demonstrate that voice cloning is feasible to some extent. However, as the findings indicate, there remains

a noticeable gap between the cloned voice and the original speaker's voice. As such, a speaker verification model can be used to detect potential voice forgeries.

