# OpenReview forum: "Efficient Speech Language Modeling via Energy Distance in Continuous Latent Space"
_NeurIPS.cc/2025/Conference — NeurIPS 2025 poster_

### Official Review · Reviewer_Fkcq · 2025-07-02

**Clarity:** 3
**Significance:** 2
**Originality:** 2
**Rating:** 4
**Confidence:** 4

**Summary:**

This paper introduces SLED, a speech language model that performs autoregressive modeling in continuous latent space using a lightweight MLP and an energy distance loss. By avoiding discretization and the hierarchical complexity associated with RVQ-based approaches like VALL-E, the method aims to simplify the modeling pipeline while preserving efficiency.
The authors claim that their approach enables efficient, streaming generation and achieves competitive results in zero-shot speech synthesis, particularly in terms of intelligibility and speaker similarity.

**Questions:**

Have you tried using codec without quantization bottleneck?

**Ethical Concerns:**

["NO or VERY MINOR ethics concerns only"]

**Final Justification:**

I am still concerned about the absence of baseline audio samples. However, since NeurIPS does not permit updating the content, I will let this pass. I have therefore increased my score to marginal accept.

**Limitations:**

Yes

**Quality:**

2

**Strengths And Weaknesses:**

Strengths:

The paper is easy to follow.

Weaknesses:
I have several concerns:
1. The use of continuous representations and autoregressive modeling is not new. The core novelty appears to be the addition of a repulsive term to the loss function, intended to better match the target distribution. However, the model does not explicitly learn a per-token distribution, making it less expressive than approaches based on flow matching, diffusion, or token-wise GMMs. In this sense, the method feels more like a variant of Tacotron than a fundamentally new approach.
2. While the authors advocate for continuous modeling, they still rely on a codec (e.g., Encodec) trained with discrete quantization. This undermines the claim that their method avoids quantization-based bottlenecks.
3. The demo page lacks baseline audio samples, making it difficult for reviewers to meaningfully compare the quality of SLED against prior work.
4. The paper appears to conflate autoregressive generation with streaming capability. In practice, autoregressive models do not inherently support streaming unless there is a mechanism to determine when to consume input incrementally and when to pause for additional context—something not clearly addressed in the current design.

---

> ### Author Rebuttal · Authors · 2025-07-30
>
> ***Thank you for your thoughtful review. We believe there may have been some misunderstandings, and we hope the following clarifications will help resolve them.***
>
> > *However, the model does not explicitly learn a per-token distribution, making it less expressive than approaches based on flow matching, diffusion, or token-wise GMMs. In this sense, the method feels more like a variant of Tacotron than a fundamentally new approach.*
>
> **Actually, our method explicitly learns a per‑token probability distribution. The repulsive term is derived from energy statistics [1, 2], which provide a principled measure of *distributional discrepancy* between model and data. This is the key difference from prior regression‑based systems such as Tacotron.** As detailed in Section 2, the generalized energy distance (Eq.4) is a special case of Maximum Mean Discrepancy (Eq.2) when the distance function is conditionally negative definite [1] (e.g. Gaussian Kernel or $d(x,y) = \lVert x - y\rVert_2^{\beta}, 0<\beta<2$). **MMD supplies a closed‑form metric for continuous distributions, analogous to the KL divergence for discrete ones. Therefore, minimizing the energy distance guides the model distribution toward the data distribution in much the same way that cross‑entropy minimization does in the discrete setting.** Moreover, modeling continuous distributions with the energy distance is theoretically more expressive than using analytically tractable distributions such as Gaussians or GMMs, because it imposes no prior assumptions on the latent‑space distribution. We elaborate on this in Appendix D.
>
> [1] Strictly proper scoring rules, prediction, and estimation.
> [2] Energy statistics: A class of statistics based on distances.
>
> > *1. While the authors advocate for continuous modeling, they still rely on a codec (e.g., Encodec) trained with discrete quantization. This undermines the claim that their method avoids quantization-based bottlenecks.*
> *2. Have you tried using codec without quantization bottleneck?*
>
> **We additionally experimented with training a VAE as a continuous tokenizer to extract latent speech representations.** As VAE has no discrete bottleneck, it can encode speech into continuous vectors at a lower frame rate, thereby reducing the number of autoregressive steps in the downstream language model. We see this as another advantage of continuous modeling. We trained a VAE with compression ratios of 2 → 4 → 5 → 5 → 8, mapping 24 kHz audio to 15 Hz, 128d vectors. We then train another SLED on top of this tokenizer.
>
> **L**: 3s Prefix as Prompt **R**: Reference Utterance as Prompt
> |WER-C|SIM|WER-C|SIM|
> |-|-|-|-|
> |1.56 | 0.551|1.53|0.676|
>
> We find, even under the lower frame rate setting, the model trained on the VAE‑based continuous tokenizer achieves comparable word accuracy while delivering better voice cloning similarity. **These findings further highlight the feasibility of speech language modeling in a continuous latent space and also show the importance of tailoring the tokenizer to continuous autoregressive models.**
>
> > *The demo page lacks baseline audio samples, making it difficult for reviewers to meaningfully compare the quality of SLED against prior work.*
>
> Thank you for your suggestion; we will improve the way we create the demo.
>
>
> > *The paper appears to conflate autoregressive generation with streaming capability. In practice, autoregressive models do not inherently support streaming unless there is a mechanism to determine when to consume input incrementally and when to pause for additional context—something not clearly addressed in the current design.*
>
> We believe there is some misunderstanding. **In Section 3.4, we actually explain our streaming mechanism**: because the text‑to‑speech task exhibits strong monotonic alignment, we design the mechanism as a fixed‑ratio text–speech interleaving. **We also specify the exact interleaving ratios we used (5:45 and 5:20) when presenting the streaming generation results in Table 2.**
>
> ***We hope these clarifications give you a clearer understanding of our work. Thank you again for your efforts in reviewing.***

---

> > ### Comment · Reviewer_Fkcq · 2025-08-07
> >
> > 1. It seems the authors may have misunderstood the concept of MMD. MMD does not learn explicit model distributions; rather, it is akin to GANs in that it learns implicit distribution.
> > 2. My second concern remains unaddressed.
> > 3. My third concern remains unaddressed.
> > 4. Monotonic alignment alone does not guarantee streamability. For streaming generation, it is essential to determine when to wait for input and when generation can proceed.

---

> ### Author Response · Authors · 2025-08-07
> **Response to Reviewer Fkcq: Clarification on "explicit"**
>
> Thank you for your response. We would like to offer the following clarification:
>
> **The term “explicit” in our rebuttal is used in exactly the same sense as in your review statement**:
> >*However, the model does not **explicitly** learn a per-token distribution, making it less expressive than approaches based on flow matching, diffusion, or token-wise GMMs.*
>
> Here, **“explicit”** in your initial review means that whether our model is a probabilistic generative model. So we use **“explicit”** in our rebuttal to claim that our model is indeed a probabilistic generative model.
>
>
>
> **What you refer to as “explicit” in the later comment:**
> >*It seems the authors may have misunderstood the concept of MMD. MMD does not learn **explicit** model distributions; rather, it is akin to GANs in that it learns implicit distribution.*
>
> In your later comment, **“explicit”** describes probabilistic generative models that do directly model the sample’s log-probability. **From this standpoint, the diffusion approaches you mention in the initial review are also NOT explicit**.

---

> ### Author Response · Authors · 2025-08-07
> **Response to Reviewer Fkcq**
>
> Thank you for your response. We would like to offer the following clarification:
>
> > *My second concern remains unaddressed: While the authors advocate for continuous modeling, they still rely on a codec (e.g., Encodec) trained with discrete quantization.*
>
>  In our rebuttal, **we do experimented with training a VAE as a continuous tokenizer to extract latent speech representations, further avoiding any quantization bottleneck.** Please refer to our rebuttal.
>
> > *My third concern remains unaddressed: The demo page lacks baseline audio samples.*
>
> According to the NeurIPS rebuttal guidelines, no anonymous URLs may be provided within it and content that has already been provided at the anonymous URL cannot be modified. We believe that **demo-design issues are not appropriate for discussion in the rebuttal.** Sorry for the inconvenience.
>
> > *Monotonic alignment alone does not guarantee streamability. For streaming generation, it is essential to determine when to wait for input and when generation can proceed.*
>
> Actually, the monotonic alignment property enables straightforward, **interleaved streaming strategy** (fixed interleaving ratio, much like wait-k-stride-n in Simultaneous MT.). This is a very popular streaming strategy in streamable TTS [1]. We kindly refer Reviewer Fkcq to Section 3.4.
>
>
> [1] CosyVoice 2: Scalable Streaming Speech Synthesis with Large Language Models

---

> ### Author Response · Authors · 2025-08-09
>
> Dear Reviewer Fkcq,
>
> Could you spare some time to look over our rebuttal and the comments? If you feel there are any concerns that remain unaddressed, could you please specify them in detail?
>
> Thank you again for your efforts in reviewing.

---

### Official Review · Reviewer_pbK8 · 2025-07-02

**Clarity:** 3
**Significance:** 3
**Originality:** 3
**Rating:** 4
**Confidence:** 3

**Summary:**

This paper introduces SLED, which is a language modeling-based speech synthesis approach based on continuous representations instead of discrete. From an architectural point of view, the SLED model is simple enough: it only requires a speech VAE as continuous encoder, a decoder only transformer, and a very lightweight generative head in AdaLN structure. The algorithm of SLED is also easy to understand: it is essentially an MSE loss between the generated latent and ground truth latent, plus a "repulsive" term that intuitively encourages randomness between two independently generated latents. This optimization target is derived from energy modeling theory. The SLED model is trained and evaluated using zero-shot TTS task, and compared to other TTS baselines in different genres. The experiments show that SLED achieves comparable performance to strong baselines but with a small parameter count and high efficiency. The authors also implemented classifier free guidance and streaming inference as bonus functions.

**Questions:**

* Why do you think the speaker similarity of SLED is relatively low?
* Continuous autoregressive generation using diffusion is suffering from sampling efficiency if the generation is frame-by-frame, but if this is  done chunk-wise instead, the sampling efficiency could not be a big problem. There are some explorations into this approach, e.g. DiTAR (https://arxiv.org/pdf/2502.03930). How do you compare SLED with those?
* Why is the classifier free guidance mechanism designed in such way? Interpolating in the z-space does not seem very intuitive to me.

**Ethical Concerns:**

["NO or VERY MINOR ethics concerns only"]

**Final Justification:**

In my original review, I raised two key questions:
* What is the fundamental difference between SLED and MELLE, since MELLE also incorporates a similar loss?
* What is the advantage of SLED over DiTAR, as the latter is a hybrid approach between AR and NAR?

In the rebuttal, the authors resolved these questions by:
* Clarifying that SLED is a theory-grounded solution instead of empirical method. With the energy modeling theory behind, the model behavior is more explainable, and more inspirations can be drawn for future work.

* Comparing the performance of SLED and DiTAR under the same setup, showing that it can reach better performance under 1/10 FLOPs of DiTAR, because of the very lightweight prediction head.

Therefore I'd like to raise the final score to 4 (borderline accept).

**Limitations:**

yes

**Quality:**

3

**Strengths And Weaknesses:**

### Strengths
The paper is theoretically solid, and the idea of exploring energy modeling in autoregressive TTS has not been researched with depth. The architecture and algorithmic design of SLED is also simple enough for others to follow. The simplicity in design also benefits parameter efficiency and inference latency, since only a small generative head upon an LLM backbone is enough. The writing of this paper is rigorous and professional.

### Weaknesses
From the final loss form, the loss still looks like an MSE for regression, plus an additional "repulsive" term. Although this looks doubtful, the authors manage to explain the importance of this repulsive term by theory and experiments. However, this brings another concern (and also mentioned in the paper) that SLED does not differ from MELL-E fundamentally, because MELL-E also introduces the penalty for adjacent frames (perhaps empirically). From the comparison in Table 1 we can also find MELL-E actually outperforms SLED in 5 out of 6 metrics. The authors also mention that "MELLE incorporates an NAR refinement, whereas SLED is purely autoregressive and can generate in a streaming manner" as an advantage of SLED over MELLE. So, how will MELLE perform without this NAR refinement compared to SLED?

In other words, I think the difference and advantage of SLED over MELLE need to be further illustrated and studied. Hence I tend to give borderline reject to this paper, but would be happy to raise it if the questions are properly addressed.

---

> ### Author Rebuttal · Authors · 2025-07-30
>
> ***Thank you for your thoughtful review. We would like to address your concerns as follows.***
>
> >*Further illustrate the difference and advantage of SLED over MELLE.*
>
> a. **The main contribution of SLED**: Our main contribution lies in **presenting a concise and theory‑grounded framework to train continuous speech language model.**
> $$
> L_t = \mathrm{E}_{h_t,h_t'}[d(h_t.h_t^*)-d(h_t,h_t')]
> $$
> where $d$ is any conditionally negative definite distance function [1,2] (e.g. Gaussian Kernel).
> This framework is grounded in the theory of maximum mean discrepancy, which quantifies the distributional discrepancy. **Optimizing this loss guarantees the model distribution to be closer to the data distribution.**
>
> b.  **We train SLED with a specific instance of a conditionally negative‑definite distance function, $d(x,y) = \lVert x - y\rVert_2^{\beta}, 0<\beta<2$. This choice also lets us clarify which heuristic loss designs proposed in earlier work are genuinely beneficial and which ones turn out to be unnecessary.** Specifically, we link the framework’s repulsive term
>
> $L_{repulsive}^{t} = - \mathrm{E}_{h_t,h_t'}[d(h_t,h_t')]$
>
> to the flux loss used in MELLE [3],
>
> $L_{flux}^{t}= -\lVert y_{t} - y_{t-1}^{*}\rVert_1$.
>
> The MELLE paper observes that their model often enters a degenerate loop of repetitive predictions, yielding poor audio quality. Therefore, MELLE heuristically forces the current frame’s prediction to stay away from the previous frame’s target. **We argue that its actuall effectiveness stems from the fact that the flux loss approximates the repulsive term in energy distance to some degreee, which makes their training closer to capture distributional difference, much like our framework, instead of pure regression.**
>
> c. **By exploiting the theoretical link between energy distance and maximum mean discrepancy, we obtain some insights that can guide future research.** The distance function should be conditionally negative definite. For the standard $L_n$ metrics, the only one that satisfies the required condition is $d(x,y) = \lVert x - y\rVert_2^{\beta}, 0<\beta<2$. This points to using an L2 distance rather than an L1 distance, and to taking the L2 distance in its square‑root form (β=1) instead of the squared form.
>
> [1] Strictly proper scoring rules, prediction, and estimation.
> [2] Energy statistics: A class of statistics based on distances.
> [3] Autoregressive Speech Synthesis without Vector Quantization.
>
>
>
>
> > *Why do you think the speaker similarity of SLED is relatively low?*
>
> We attribute the slightly lower speaker similarity scores of SLED reported in the manuscript to our use of EnCodec as the latent encoder. In our initial choice of tokenizer, to keep tokenizer effects out of the discrete‑vs‑continuous comparison, we adopted the same tokenizer setup as VALL‑E, employing the same 8‑VQ-layer EnCodec tokenizer.
>
> However, we additionally experimented with training a VAE as a tokenizer to extract continuous latent representations. As VAE has no discrete bottleneck, it can encode speech into continuous vectors at a lower frame rate, thereby reducing the number of autoregressive steps in the downstream language model. We see this as another advantage of continuous modeling. We trained a VAE with compression ratios of 2 → 4 → 5 → 5 → 8, mapping 24 kHz audio to 15 Hz, 128d vectors. We then train another SLED on top of this tokenizer.
>
> **L**: 3s Prefix as Prompt **R**: Reference Utterance as Prompt
> |WER-C|SIM|WER-C|SIM|
> |-|-|-|-|
> |1.56 | 0.551|1.53|0.676|
>
> We find the model trained on the VAE‑based continuous tokenizer achieves better voice cloning similarity even under the lower frame rate setting. These findings show the importance of tailoring the tokenizer to continuous autoregressive models.
>
>
> > *Continuous autoregressive generation using diffusion is suffering from sampling efficiency if the generation is frame-by-frame, but if this is done chunk-wise instead, the sampling efficiency could not be a big problem. There are some explorations into this approach, e.g. DiTAR (https://arxiv.org/pdf/2502.03930). How do you compare SLED with those?*
>
> We see SLED and DiTAR as two distinct routes to improve the efficiency of continuous autoregressive speech synthesis. DiTAR takes a **semi‑autoregressive** approach: by trimming the number of sequential autoregressive steps, it gives a time complexity that sits between fully AR and fully NAR. In contrast, SLED tackles the problem by **boosting the efficiency of each individual autoregressive step**. We compare the perfomrance and efficiency of both methods trained on 50k hrs Libriheavy/LibriLight in the following table: (In this table, we report the results of SLED trained on Encodec's latent representation to keep consistent with our manuscript.)
>
> |     | WER | SIM | #Params | #Transformer Layer | RTF under Batch Size 1 | FLOPs|
> |-|-|-|-|-|-|-|
> |SLED | 1.51 | 0.664 | 0.2B | 12 | 0.8 | 280G |
> |DiTAR (patch size 4)| 1.78 | 0.64 | 0.6B |6+36+6 | 0.66 | 2750G |
>
> In our opinions, the SLED architecture may offer the following advantages over DiTAR:
>
> a. DiTAR is actually a hierachical model. Although patching cuts down the number of autoregressive steps, the per‑step generation workload in the DiTAR model is heavy, therefore each step relies on a relatively large local decoder instead of lightweight MLP in SLED. This will offset some of the advantages brought by the patch. Moreover, iterating within a local DiT model can make it prone to losing global information. In fact, Table 5 of the DiTAR paper notes that the local DiT model must incorporate historical‑patch dependencies to generate correctly; this further increases the model’s complexity and negates the computational advantages gained from patching. As a result, DiTAR attains an RTF comparable to SLED, but it incurs nearly ten times the FLOPs (though DiTAR is 3 times larger than SLED).
>
> b. From the point of reducing autoregressive steps, we think a more promising route is to train a continuous tokenizer that compresses the frame rate, thereby reducing the number of steps the autoregressive model must take. Our experiments with a VAE-based continuous tokenizer show that a well‑designed tokenizer that lowers the frame rate can preserve comparable word‑level accuracy while even boosting cloning similarity. In our view, pairing this strategy with SLED’s modeling scheme is the more efficient choice.
>
>
>
> > *Why is the classifier free guidance mechanism designed in such way? Interpolating in the z-space does not seem very intuitive to me.*
>
> We devised the CFG approach in SLED following the rationale below.
> In a probabilistic model, CFG is essentially doing the following: $p^{\text{sample}}(x \mid c) \propto p(x \mid c) \left[\frac{p(x \mid c)}{p(x)} \right]^{s}$. Let $l=logp$, then we have the interpolating rule $l^{sample}
> =l^{c} + s \bigl(l^{c} - l^{u}\bigr)$. In a discrete AR model, $l$ represents the model’s logits, which is the hidden states before the softmax is applied. In SLED, the lightweight MLP plays a role analogous to the softmax layer in discrete models (Both transforming an autoregressive model’s encoded hidden states into a probability distribution). Therefore, we treat the hidden states $z$ in SLED like the logits $l$ in a discrete AR model and apply the same interpolation strategy during CFG generation.
>
> ***We hope these clarifications give you a clearer understanding of our work. Thank you again for your efforts in reviewing.***

---

> > ### Comment · Reviewer_pbK8 · 2025-08-05
> >
> > Thank very much to the authors for this detailed reply. My questions are resolved now, and I suggest the authors to include the new experiments (especially the comparison with DiTAR as its hybrid approach is of great interest to many people) if this paper is accepted.

---

### Official Review · Reviewer_BhHp · 2025-07-03

**Clarity:** 3
**Significance:** 3
**Originality:** 4
**Rating:** 5
**Confidence:** 4

**Summary:**

This paper introduces SLED, a novel approach to speech language modeling that operates in a continuous latent space using energy distance. By avoiding discretization via residual vector quantization (RVQ) and complex hierarchical architectures, SLED simplifies the modeling pipeline while preserving speech information richness. It employs a lightweight implicit conditional generative module trained with generalized energy distance (GED), a form of maximum mean discrepancy (MMD), enabling efficient autoregressive modeling. Experimental results demonstrate strong performance in zero-shot and streaming speech synthesis, outperforming discrete counterparts like VALL-E in parameter efficiency and quality metrics (e.g., WER, SIM, DNSMOS)

**Questions:**

1. Are there any experimental results using more Encodec RVQ layers?
2. How does the model’s performance scale with larger parameter sizes  and extended training data beyond 50,000 hours (like in Emilia)?

**Ethical Concerns:**

["NO or VERY MINOR ethics concerns only"]

**Final Justification:**

All my concerns are resolved

**Limitations:**

yes

**Quality:**

3

**Strengths And Weaknesses:**

Strengths
* SLED eliminates discretization errors and information loss inherent in RVQ-based methods, leveraging continuous representations for more faithful speech reconstruction.
* By eschewing hierarchical structures required for multi-stream RVQ tokens, SLED reduces complexity while maintaining efficiency in both training and inference.
* The use of GED as a strictly proper scoring rule enables stable training, and the absence of iterative sampling or post-AR refinement enhances efficiency for long speech sequences.

Weakness
* The main drawback is that the selected baselines exhibit weak performance.

---

> ### Author Rebuttal · Authors · 2025-07-30
>
> ***Thank you for your acknowledgement of this work. We would like to address your remaining concerns as follows.***
>
> > *Are there any experimental results using more Encodec RVQ layers?*
>
> In our initial experiments with Encodec, we constructed the latent representation by summing the embeddings from first eight VQ layers. The goal was to align with VALL‑E’s setup and remove any confounding effects from the tokenizer itself.
> In later experiments, to further showcase the benefits of modeling in a continuous latent space, we explored two modifications to the tokenizer:
>
> **1.Increase the number of Encodec RVQ layers used to build the continuous latent representation.**
>
> **2.Retrain a continuous tokenizer based on a VAE architecture, which contains no discrete bottleneck at all.**
>
> In the first experiment, we evaluated configurations with $n_q=16$ and $n_q=32$. The $n_q=16$ setup performed on par with $n_q=8$, but training collapsed at $n_q=32$ and the model failed to generate intelligible speech. We attribute this collapse to the excessive number of codebooks: **when too many are combined, the RVQ‑VAE effectively degenerates into a plain auto‑encoder, and the summed embeddings no longer carries any meaningful structural priors. Such an unstructured latent space hampers continuous‑space modelling.** In contrast, a moderate number of codebooks achieves a useful compromise between information preservation and latent‑space regularisation—functionally similar to the KL term in a VAE—which in turn supports autoregressive modelling in the continuous latent space.
>
> In the second experiment, we experimented with training a VAE as a continuous tokenizer to extract latent representations. As VAE has no discrete bottleneck, it can encode speech into continuous vectors at a lower frame rate, thereby reducing the number of autoregressive steps in the downstream language model. We see this as another advantage of continuous modeling. We trained a VAE with compression ratios of 2 → 4 → 5 → 5 → 8, mapping 24 kHz audio to 15 Hz, 128d vectors. We then train another SLED on top of this tokenizer.
>
> **L**: 3s Prefix as Prompt **R**: Reference Utterance as Prompt
> |WER-C|SIM|WER-C|SIM|
> |-|-|-|-|
> |1.56 | 0.551|1.53|0.676|
>
> We find, even under the lower frame rate setting, the model trained on the VAE‑based continuous tokenizer achieves comparable word accuracy while delivering better voice cloning similarity. These findings further highlight the feasibility of speech language modeling in a continuous latent space and also show the importance of tailoring the tokenizer to continuous autoregressive models.
>
> > *How does the model’s performance scale with larger parameter sizes and extended training data beyond 50,000 hours (like in Emilia)?*
>
> Currently we have scaled the model to 0.5 billion parameters by initializing the AR backbone from Qwen2‑0.5B, but we observed no clear improvement. More complex speech‑generation tasks (e.g., InstructTTS) may demand stronger text understanding, in which case initialization from a pre‑trained LLM could be more advantageous. We plan to investigate the scaling laws of continuous speech‑language models in future work.
>
> ***We hope these clarifications give you a clearer understanding of our work. Thank you again for your efforts in reviewing.***

---

> > ### Comment · Reviewer_BhHp · 2025-08-05
> >
> > Thank you for your response. My concerns are resolved

---

### Official Review · Reviewer_GWrt · 2025-07-06

**Clarity:** 3
**Significance:** 3
**Originality:** 3
**Rating:** 4
**Confidence:** 3

**Summary:**

This paper proposes SLED, a method for modeling speech auto-regressively in a continuous space using generalized energy distance as the training objective. By operating on continuous speech features rather than discrete tokens, SLED preserves finer-grained details and offers greater computational efficiency compared to multi-stream discrete representations. Experiments on LibriSpeech demonstrate that SLED achieves competitive performance with baseline methods in synthesis quality, and ablation studies highlight the importance of the repulsive term in the energy distance formulation.

**Questions:**

1. How is the reference utterance used as a prompt within the SLED architecture?
2. Have the authors considered fine-tuning from a pre-trained language model? Would initializing from a pre-trained LLM improve convergence or synthesis quality?

**Ethical Concerns:**

["NO or VERY MINOR ethics concerns only"]

**Final Justification:**

My primary concern was the reliance on Encodec features; however, the authors have demonstrated that SLED also works with VAE-based features, confirming that the method is indeed generalizable. I still have a minor reservation about its robustness to varying speaking rates, but this is not a significant issue.

Overall, I will maintain my positive score.

**Limitations:**

See Strengths And Weaknesses.

**Paper Formatting Concerns:**

None.

**Quality:**

3

**Strengths And Weaknesses:**

# Strength

1. The proposed method offers an elegant approach to modeling sequences of continuous speech features in an auto-regressive manner.
2. Experiments on LibriSpeech demonstrate that SLED achieves competitive or superior synthesis quality using purely auto-regressive generation on continuous features.
3. The model supports streaming text-to-speech (TTS) with only a slight degradation in output quality.

# Weakness

1. SLED relies on a pre-trained codec model to produce continuous embeddings. It would be helpful to compare this with alternatives such as mel-spectrograms or self-supervised features like WavLM.
2. While the paper claims improved efficiency over multi-stream discrete token modeling, it lacks a thorough evaluation of computational cost. Additionally, classifier-free guidance introduces extra overhead that should be accounted for.
3. The streaming policy assumes a fixed text-to-speech ratio, which may not generalize well to varied speaking rates. Moreover, the paper should clarify the exact latency—i.e., the delay between receiving a text token and producing its corresponding audio.

---

> ### Author Rebuttal · Authors · 2025-07-30
>
> ***Thank you for your thoughtful review. We would like to address your concerns as follows.***
>
> > *SLED relies on a pre-trained codec model to produce continuous embeddings. It would be helpful to compare this with alternatives such as mel-spectrograms or self-supervised features like WavLM.*
>
> Thank you for your suggestions. We additionally experimented with training a VAE as a continuous tokenizer to extract latent speech representations. As VAE has no discrete bottleneck, it can encode speech into continuous vectors at a lower frame rate, thereby reducing the number of autoregressive steps in the downstream language model. We see this as another advantage of continuous modeling. We trained a VAE with compression ratios of 2 → 4 → 5 → 5 → 8, mapping 24 kHz audio to 15 Hz, 128d vectors. We then train another SLED on top of this tokenizer.
>
> **L**: 3s Prefix as Prompt **R**: Reference Utterance as Prompt
> |WER-C|SIM|WER-C|SIM|
> |-|-|-|-|
> |1.56 | 0.551|1.53|0.676|
>
> We find, even under the lower frame rate setting, the model trained on the VAE‑based continuous tokenizer achieves comparable word accuracy while delivering better voice cloning similarity. These findings further highlight the feasibility of speech language modeling in a continuous latent space and also show the importance of tailoring the tokenizer to continuous autoregressive models.
>
> > *While the paper claims improved efficiency over multi-stream discrete token modeling, it lacks a thorough evaluation of computational cost. Additionally, classifier-free guidance introduces extra overhead that should be accounted for.*
>
> Sorry for the confusion. We actually include an efficiency analysis in Appendix E; for your convenience, we reproduce the table here:
>
> *Table 5. Inference latency and RTF of generating a 10 s speech
> \* We use teacher‑forcing‑style AR forward pass to estimate FLOPs in the AR module.*
>
> | Model | AR Latency | Gen. Module Latency | NAR Latency | RTF | FLOPs\* |
> |-------|-----------:|--------------------:|------------:|----:|--------:|
> | VALL‑E | 6.91 s | 0.16 s (Softmax) | 0.94 s | 0.8 | 8 × 222.7 G |
> | SLED (No CFG) | 6.82 s | 1.23 s (MLP) | — | 0.8 | 280.05 G |
>
>
> This table compares the computational efficiency of continuous and discrete modeling approaches when both use the same tokenizer (Encodec). the SLED figures shown are obtained without CFG. When CFG is enabled, the computational load (FLOPs) of the global AR component doubles (while latency is almost unaffected), whereas the MLP component remains unchanged. We recognize that the additional computational overhead introduced by CFG remains an open issue. In future work, we intend to investigate guidance‑free approaches for energy distance-based continuous autoregressive modeling to alleviate this overhead.
>
> > *The streaming policy assumes a fixed text-to-speech ratio, which may not generalize well to varied speaking rates. Moreover, the paper should clarify the exact latency—i.e., the delay between receiving a text token and producing its corresponding audio.*
>
> In our opinion, a fixed text‑to‑speech ratio does not hurt the model’s ability to generalize to different speaking rates, because the audio at a given position does not have to contain words that exactly match the current text segment. The streaming synthesis section of the demo at the anonymous URL showcases samples generated at various speaking rates.
>
> We also measured the latency between receiving the first text chunk (first five tokens) and emitting its corresponding 0.6‑second audio segment. The delay is about 0.3–0.4 s, though it can vary with hardware. To ensure full reproducibility, we have included the complete training and inference code in the supplementary materials and will release it publicly.
>
> > *How is the reference utterance used as a prompt within the SLED architecture?*
>
> We prepend the model input with "prompt text + input text + prompt speech", enabling the model to generate speech whose timbre closely matches that of the prompt speech.
>
> > *Have the authors considered fine-tuning from a pre-trained language model? Would initializing from a pre-trained LLM improve convergence or synthesis quality?*
>
> We have experimented with initializing from Qwen2-0.5B but observed no clear improvement. In our view, TTS largely hinges on the model’s ability to generate speech, which gains little from a text-only pre-trained model. However, more complex speech generation tasks (e.g., InstructTTS) may require stronger text understanding, in which case initializing from a pre‑trained LLM could be more beneficial.
>
> ***We hope these clarifications give you a clearer understanding of our work. Thank you again for your efforts in reviewing.***

---

> > ### Comment · Reviewer_GWrt · 2025-08-03
> >
> > Thank you for the response. My concerns are resolved.

---

### Decision · Program_Chairs · 2025-09-17

**Decision:**

Accept (poster)

**Comment:**

This paper presents SLED, an autoregressive speech synthesis method operating on continuous latent representations and trained with generalized energy distance (GED). SLED preserves fine-grained speech details, supports streaming (monotonic!) TTS, and achieves competitive zero-shot and genre-diverse synthesis with high efficiency compared to baselines like VALL-E and DiTAR. Strengths, as highlithted by the reviewers, include theoretical grounding, simplicity, parameter efficiency, and clear exposition. During the discussion period, the authors addressed concerns about similarity to MELL-E and show superior FLOPs efficiency over DiTAR. Weaknesses are minor, including reliance on pre-trained encoders, limited large-scale evaluation, and questions about robustness to varying speaking rates (but this is due to the choice of fixed streaming policy, which isn't the main point of this paper). Overall, SLED offers a meaningful methodological contribution with strong empirical support.